

# Abnormal expression of *HOXD11* promotes the malignant behavior of glioma cells and leads to poor prognosis of glioma patients

Jialin Wang[1,2,*], Zhendong Liu[3,*], Cheng Zhang[4], Hongbo Wang[5], Ang Li[3], Binfeng Liu[1], Xiaoyu Lian[1], Zhishuai Ren[1], Wang Zhang[6], Yanbiao Wang[1], Bo Zhang[5], Bo Pang[7] and Yanzheng Gao[3]

[1] Zhengzhou University People's Hospital, Henan Provincial People's Hospital, Zhengzhou, Henan, China
[2] Department of Microbiome Laboratory, Henan Provincial People's Hospital, People's Hospital of Zhengzhou University, Zhengzhou University, Zhengzhou, Henan, China
[3] Department of Surgery of Spine and Spinal Cord, Henan Provincial People's Hospital, People's Hospital of Zhengzhou University, School of Clinical Medicine, Henan University, Zhengzhou, Henan, China
[4] North Broward Preparatory School, Nord Anglia Education, Coconut Creek, FL, United States of America
[5] Henan University People's Hospital, Henan Provincial People's Hospital, Zhengzhou, Henan, China
[6] Department of Neurosurgery of the First Affiliate Hospital of Harbin Medical University, Harbin, Heilongjiang, China
[7] Department of Neurosurgery, The Fourth Medical Center of Chinese PLA General Hospital, Beijing, China
[*] These authors contributed equally to this work.

Corresponding author
Yanzheng Gao,
yanzhenggaohn@163.com

## ABSTRACT

**Background**. Homeobox D11 (*HOXD11*) plays an important role in a variety of cancers, but its precise role in gliomas remains unclear. This study aimed to explore the relationship between *HOXD11* and gliomas by combining bioinformatics methods with basic experimental validation.

**Materials and methods**. Obtain gene expression information and clinical information of glioma and non-tumor brain tissue samples from multiple public databases such as TCGA (666 glioma samples), CGGA (749 glioma samples), GEPIA(163 glioblastoma samples and 207 normal control samples), GEO (GSE4290 and GSE15824). Nine cases of glioma tissue and five cases of normal control brain tissue were collected from the clinical department of Henan Provincial People's Hospital for further verification. A series of bioinformatic analysis methods were used to confirm the relationship between *HOXD11* expression and overall survival and clinical molecular characteristics of patients with glioma. RT-qPCR was used to verify the change of expression level of *HOXD11* in glioma cells and tissues. MTT assay, colony formation assay, wound-healing assay, immunofluorescence staining, flow cytometry and western blotting were used to detect the effect of *HOXD11* on the biological behavior of glioma cell line U251.

**Results**. The high expression of *HOXD11* was significantly related to age, World Health Organization (WHO) grade, chemotherapy status, histological type, and even 1p19q codeletion data and isocitrate dehydrogenase (IDH) mutation. *HOXD11*, as an independent risk factor, reduces the overall survival of glioma patients and has diagnostic value for the prognosis of glioma. Gene Set Enrichment Analysis (GSEA) showed that *HOXD11* was significantly enriched in cell signaling pathway such as cell cycle, DNA replication and so on. Finally, we confirmed that the knockout of *HOXD11*

can inhibit the proliferation and invasion of U251 glioma cells, and change the biological behavior of tumor cells by preventing the progression of cell cycle.

**Conclusions**. *HOXD11* may be used as a candidate biomarker for the clinical application of targeted drug and prognostic assessment treatment of glioma. In addition, This study will help to explore the pathological mechanism of glioma.

## INTRODUCTION

Gliomas are one of the most common primary intracranial neoplasms, accounting for 81% of intracranial neoplasms (*Ostrom et al., 2014*). The treatment options for glioma are mainly focused on maximizing surgical removal of the tumor tissue and assisting in comprehensive radiotherapy and chemotherapy (*Taal, Bromberg & Van den Bent, 2015*). The postoperative recurrence and poor prognosis of gliomas remain a major problem. Therefore, scientists have adopted new treatment methods, such as immunotherapy and photodynamic therapy, to overcome current limitations (*Kong, Wang & Ma, 2018*; *Zavadskaya capital Te, 2015*). However, these new treatments have not yet completely improved the prognosis of gliomas. Part of the reason for this may be that the pathogenesis of glioma has not been fully elucidated.

In recent years, many studies have reported that epigenetics plays a crucial role in the development and regulation of cancer (*Dawson & Kouzarides, 2012*). Epigenetics is involved in regulating the pathophysiological process of tumors. Epigenetics is a general term for various modifications that do not change the sequence of genetic material but can lead to changes in gene expression, such as methylation, acetylation and phosphorylation (*Abbaoui et al., 2017*; *Jablonska & Reszka, 2017*; *Kondo, Shinjo & Katsushima, 2017*; *Zhang et al., 2017*). Significant advances have been made in the field of epigenetics in recent years, mainly due to the rapid development of high-throughput whole-genomes sequencing that provide researchers with all genetic expression changes associated with genetic modification (*Kim et al., 2016*).

TCGA is a widely used public database storing the genetic and clinical information of a variety of human malignant tumors, and it has made great contributions to the research on human malignant tumors (*Tomczak, Czerwinska & Wiznerowicz, 2015*). In addition, CGGA is a free database dedicated to glioma research. It also stores a large sample of gene expression profile information and patient-related clinical information, which may help researchers explore the relationship between genetic information of gene expression and clinical data related to glioma. Therefore, researchers' understanding of CGGA-based glioma is gradually expanding (*Liang et al., 2019*; *Wang et al., 2018*). For example, *Feng et al. (2019)* found that high expression of major histocompatibility complex, class I, F (HLA-F) can predict poor prognosis in patients with glioma. *Guan et al. (2018)* showed the abnormally high expression of CKLF like MARVEL Transmembrane domain containing 6 (CMTM6) is involved in the pathophysiological process to promote poor prognosis

of glioma. Therefore, our study attempts to find a biomarker related to the prognosis of glioma through the combination of bioinformatics method and traditional experimental verification.

*HOXD11* belongs to the HOX gene family, which encodes transcription factors that regulate various physiological processes. In recent years, many articles have reported that *HOXD11* plays a vital regulatory role in tumors like laryngeal squamous cell carcinoma, ovarian cancers, head and neck cancer, and others (*Cai et al., 2007*; *De Barros et al., 2016*; *Sharpe et al., 2014*). However, as far as we know, there is no report on the relationship between *HOXD11* and glioma. Here, we examined the relationship between high or low expression level of *HOXD11* and the prognosis and clinical features of glioma using bioinformatics approaches. Further indirect analysis by Gene Set Enrichment Analysis (GSEA) was used to reveal the mechanism of action of *HOXD11* in gliomas. Finally, a variety of experimental methods have been used to confirm that *HOXD11* can promote the malignant behavior of glioma cells as an oncogene by participating in the regulation of cell cycle signaling pathways. Thus, we believe this study will provide a new diagnostic biomarker for the diagnosis and therapeutic target for gliomas.

## MATERIAL AND METHODS

### Data collection

GEPIA is an integrated database platform, which is characterized by the fusion of gene sequencing data of tumor samples from TCGA database and gene sequencing data of normal control group sample tissues from The Genotype-Tissue Expression (GTEx) (*Tang et al., 2017*). This advantage of GEPIA makes up for the shortage of normal control samples in TCGA database. The expression levels of *HOXD11* in various tumors and tissues, including 163 glioblastoma, 518 low grade glioma and 207 normal brain tissue samples, were obtained from the GEPIA database. To further determine the expression level changes of *HOXD11* in glioma, we obtained gene expression data in the form of FPKM (Fragments Per Kilobase per Million) from a glioma tissue microarray (GSE4290) and a glioma cell microarray (GSE15824) from GEO database https://www.ncbi.nlm.nih.gov/geo/ for further analysis. GSE4290 contains 23 non-tumor brain tissue samples and 77 glioma tissue samples (*Sun et al., 2006*). GSE15824 contains three human-derived astrocyte (HA) cell line samples and two LN319, two LN229, two BS149 and two LN018 glioblastoma cell line samples (*Grzmil et al., 2011*).

CGGA (http://www.cgga.org.cn/) is a public database focused on glioma research, which contains many types of data and corresponding clinical information (*Guan et al., 2018*). This part of the data contains a variety of clinical information of the patients and was used to further depth analyze the relationship between changes in *HOXD11* expression level and glioma prognosis. We examined two RNA-seq datasets containing 693 and 325 samples. After excluding the data of patients with incomplete clinical data, we obtained 749 glioma samples for further analysis of various types.

In order to verify the effect of *HOXD11* on the prognosis and diagnostic value of glioma, we searched and obtained one RNA-seq dataset containing 666 glioma tissues in the

TCGA database (https://portal.gdc.cancer.gov/) and obtained clinical information of the corresponding patients.

Nine glioma tissue samples and five non-tumor brain tissue samples were collected and stored in a liquid nitrogen environment during surgery and then transferred to a -−80 °C freezer until use. Clinical samples were obtained after receiving written informed consent from patients in the clinical department of Henan Provincial People's Hospital. This study has been approved by the Institutional Review Board of Zhengzhou University.

## GSEA analysis of *HOXD11*

GSEA is widely used as a bioinformatics analysis tool for gene function annotation and analysis. The RNA-seq data obtained from the CGGA database was batch-corrected and normalized by limma packages and then classified into "group H" high expression group or "group L" low expression group based on the median expression level of *HOXD11*. Functional enrichment analysis was performed using the GSEA 4.0.2 jar software. The number of permutations was set to 1000 times, and the gene set database was set to the Kyoto Encyclopedia of Genes and Genomes (KEGG) cell signaling pathway.

## Cell culture

Human glioma cell lines (U251, LN229 and T98) and human-derived astrocyte (HA) were purchased from the Cell Bank of the Chinese Academy of Sciences (Shanghai, China). All cells were grown in incubators at 37 °C and 5% carbon dioxide and cultured using DMEM medium (Thermo Fisher Scientific, USA) plus 10% FBS (Thermo Fisher Scientific, USA).

## RNA extraction and quantitative reverse transcription polymerase chain reaction (RT-qPCR) analysis

9 glioma tissue and 5 peritumoral tissue brain tissue samples were used to detect the expression level of *HOXD11* in tissues by RT-qPCR. Three glioma cell lines (T98, U251, LN229) and one human astrocyte were used to detect the expression level of *HOXD11* in glioma cells. First, total RNA was extracted from glioma and normal brain tissue samples using Tri-Reagent (Sigma, USA). The quality and quantity assessments of total RNA were determined using NanoDrop One spectrophotometer (Thermo Fisher Scientific, USA), evaluating 260/280 nm absorbance values. Thereafter, the cDNA was reverse transcribed from the total RNA using the Transcriptor First Strand cDNA Synthesis kit (Roche, USA). RT-qPCR was performed following the guidelines for FastStart Universal SYBR Green Master (ROX) (Roche, Germany). Results were quantified using QuantStudio software (Thermo Fisher Scientific, USA) following the manufacturer's instructions. GADPH was used as an internal reference. The primer sequences used for *HOXD11* were 5′-GAACGACTTTGACGAGTGCG-3′(F) and 5′-ACGGTTGGGAAAGGAACGAA-3′(R). The expression level of *HOXD11* was measured by the "$2-\Delta\Delta CT$" method. Statistical differences between the two groups were analyzed by unpaired t-test, and results were considered statistically significant when the *p* value was < 0.05.

## Cell transfection

The small interfere RNA (siRNA) that specifically targeting *HOXD11* was purchased from Genepharma (Shanghai, China). The siRNA sequence was sense (5′-GACUUCAACUCUCUCGGAUTT-3′) and antisense(5′-AUCCGAGAGUUGAAGUCTT-3′). The negative control (NC) RNAi sequence was sense(5′-UUCUCCGAACGUGUCACGUTT-3′) and antisense, (5′-ACGUGACACGUUCGGAGAATT-3′). Cells were transfected using Lipofectamine 2000 reagent (Invitrogen, Carlsbad, CA, USA). Detection of transfection efficiency by RT-qPCR at 48 h after transfection.

## Cell proliferation assay

To detect the effect of *HOXD11* on the proliferation ability of U251 cells, MTT experiments were conducted. The transfected cells were transferred into 96-well plates ($1 \times 10^3$ cells/well). Cell proliferation ability was measured at 12 h, 24 h, 48 h and 96 h after transfection. The specific method was to add 20 ul of MTT (5mg/ml) to each well for culture. The liquid was discarded after incubation for 4 h and 150 ul DMSO was added to each well. After 15 min of shaking, the optical density (OD) of 490 nm was detected using microplate reader.

## Colony formation assay

In colony formation assay, U251 cells were planted in six-well plates at a density of about 300 cells/well. After 10 days of culture, the cells were fixed with 4% paraformaldehyde and stained with 0.1% crystal violet. Colonies larger than one mm in diameter were counted using ImageJ Software (v1.8.0) after photographing.

## Wound-healing assay

The exponential growth cells were seeded on the 6-well plate with a fusion degree of 90%. A straight linear scratch was made on the center of the plate with a pipette (200 uL). PBS was washed gently, and serum-free medium was replaced for culture. After that, the scratch width was observed and photographed under the microscope for 24 h and 48 h respectively, then was measured by ImageJ Software (v1.8.0).

## Immunofluorescence staining

Firstly, treated cells were fixed with 4% paraformaldehyde for 30 min, and then 0.3% Triton X-100 was used to penetrate for 20 min at room temperature. After blocking with 5% BSA, primary antibody Ki67 (1:1000, Abcam, UK) was added and placed in a 4 °C refrigerator overnight. After that, washed 3 times with PBS for 5 min each time. Secondary antibody was added and incubated for one hour at room temperature in the dark. After washing with PBS, DAPI was added for nucleation. The photographs were finally observed using a fluorescence microscope. After that, the total number of cells and the number of KI67 positive cells were counted by ImageJ Software (v1.8.0).

## Flow cytometry

U251 cells were digested and centrifuged to collect the cells after transfection. Precooled 75% ethanol was added, fixed overnight at −20 °C Each tube was added with 500 uL

propidium iodide (PI) dye solution, stained for 30 min in dark, and filtered through a filter. Cell cycle distribution was analyzed by flow cytometry (Canto plus, BD, USA).

## Western blotting

Total protein of transfected U251 cells was extracted using RIPA buffer with protease inhibitor. Separated by 10% SDS-PAGE gel electrophoresis, transferred to polypropylene fluoride membrane (PVDF) treated with methanol for 2 h, transferred to a shaker, sealed with TBS-T blocking solution containing 5% skimmed milk powder for 1 h. Then, the primary antibodies GAPDH (1:5000, Abcam, UK), CCNB1 (1:500, Abcam, UK), CDK1 (1:500, Abcam, UK) and Cdc25C (1:500, Abcam, UK) were added to incubate overnight at 4 °C. Washed by TBS-T 5 times for 5 min each time, secondary antibody (1:5000) was added to incubate for 1 h at room temperature, Washed by TBS-T 5 times for 5 min each time, and finally ECL color developing solution was added for color development. The developed bands were subjected to gray level analysis by ImageJ Software (v1.8.0), The difference in protein expression levels between the *HOXD11*-NC group and the *HOXD11*-siRNA group is represented by the gray value of the corresponding protein bands.

## Statistical analysis

R software (v.3.6.1 version) and GraphPad Prism (v8.0.2) was used to process data and perform statistical analysis. Unpaired $t$-test was used for comparison between the two groups. The expression levels of *HOXD11* were examined in glioma and non-tumor brain tissue samples by the Wilcox test method. The relationship between the Overall Survival rate of patients and the expression level of *HOXD11* was analyzed by the Kaplan–Meier method and Cox regression analysis, and the survival curve was drawn accordingly. The relationship between clinically relevant patient information and the expression level of *HOXD11* was analyzed using the Wilcox and Kruskal tests.

# RESULTS

## Abnormally high expression of *HOXD11* in various tumors including gliomas

Through GEPIA database, we found that *HOXD11* was abnormally highly expressed in a variety of tumors, such as ESCA, GBM, HNSC and HUSC, especially in GBM, as shown in Fig. 1A. To confirm the expression level of *HOXD11* in gliomas relative to normal controls, data from more sources were analyzed. By analyzing the data of glioma tissues and cell lines microarrays obtained from GEO database, it was found that *HOXD11* was highly expressed in both glioma tissues and cell lines, as shown in Figs. 1B and 1C. To validate the above analysis results, we performed RT-qPCR at tissue and cell levels, respectively. The results showed that *HOXD11* was highly expressed in glioma tissues and cell lines compared with normal brain tissues and human astrocytes, as shown in Figs. 1D and 1E. Among them, the relevant clinical information of 9 glioma tissue samples was shown in Table S1.

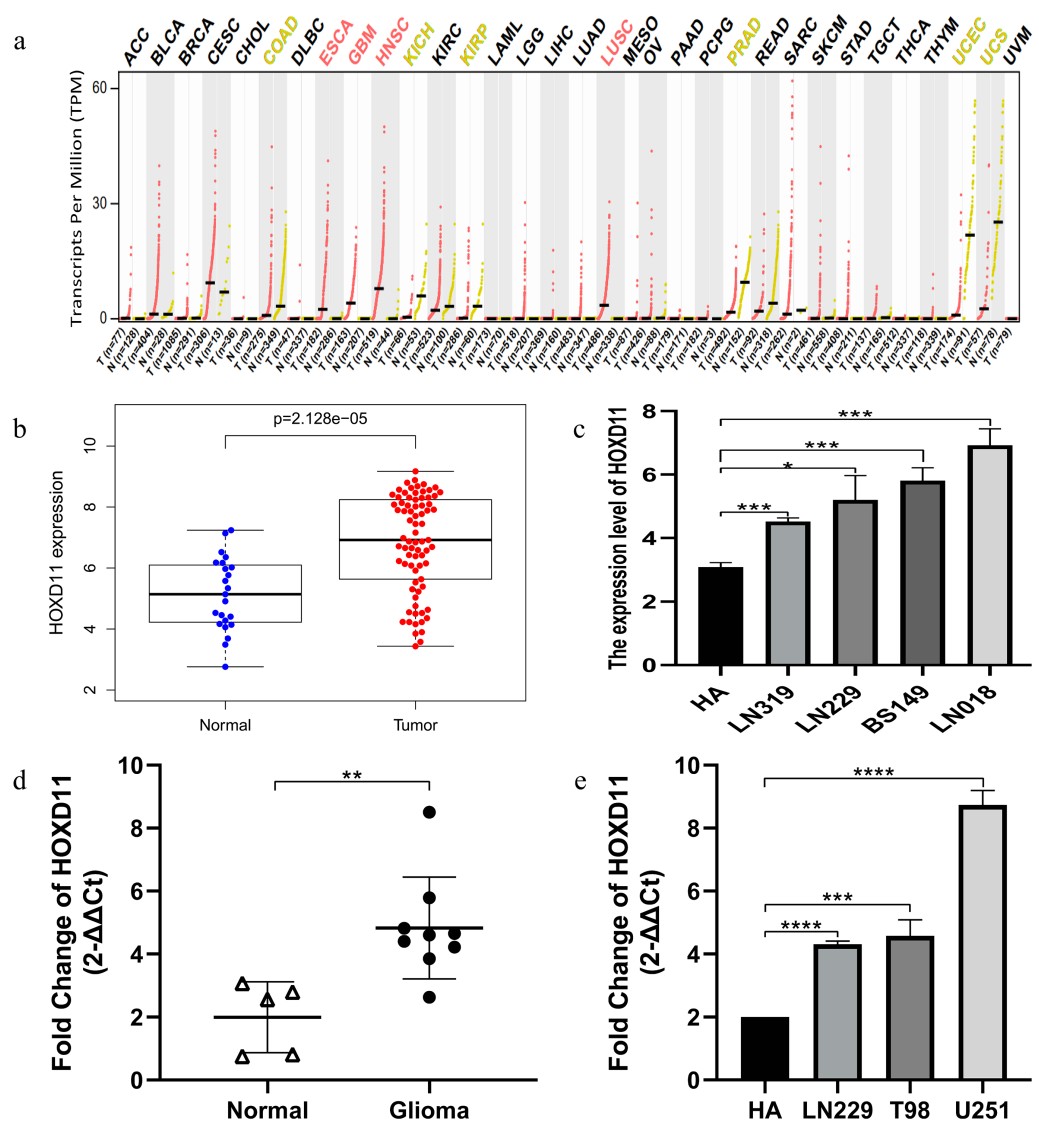

**Figure 1** **The higher expression level of *HOXD11* in glioma tissues and cells.** (A) *HOXD11* was highly expressed in glioblastoma (GBM) tissues($n = 163$) relative to normal brain tissues ($n = 207$) from GEPIA database; (B) The expression level of *HOXD11* in glioma tissues ($n = 23$) and non-tumor brain tissues ($n = 77$) in GSE4290; (C) The expression level of *HOXD11* in glioma cell lines ($n = 2$ in LN319, $n = 2$ in LN229, $n = 2$ in BS149 and $n = 2$ in LN018) and human-derived astrocyte ($n = 3$ in HA) in GSE15824; (D) RT-qPCR in glioma tissues ($n = 9$) and non-tumor brain tissues ($n = 5$) collected from clinical; (E) RT-qPCR in glioma cell lines (LN229,T98 and U251) and human-derived astrocyte (HA). (C–E) These graphs show the difference in the expression of *HOXD11* at the mRNA level, and are represented by *($p < 0.05$), **($p < 0.01$), ***($p < 0.001$) or ****($p < 0.001$) Statistical differences determined by one-way ANOVA.

## Correlation between the expression level of *HOXD11* and the clinical and prognosis of patients with glioma

To clarify the relationship between high expression of *HOXD11* and the clinical and prognosis of glioma patients, we conducted further studies using glioma sequencing data

and relevant clinical information from CGGA database. A total of 749 glioma samples with complete clinical information were obtained from the CGGA database, which contains a variety of clinical information such as Primary-Recurrent-Secondary (PRS) type, age (age at diagnosis), gender, histological type, and chemotherapy status, as well as 1p19q codeletion status and isocitrate IDH mutation status. More detailed information of clinical characteristics is shown in Table S2.

The Wilcox and Kruskal tests were used to analyze the relationship between clinically relevant patient information and the expression level of *HOXD11*. The expression level of *HOXD11* was significantly correlated with PRS type, WHO grade, chemotherapy status, age, 1p19q codeletion data, IDH mutation status and histological type (Figs. 2A–2G). The expression level of *HOXD11* increases as the WHO grade and age of the glioma increases ($p < 0.001$). As for the IDH mutation status, there was a higher level of gene expression in wildtype than in the mutant ($p < 0.001$). Moreover, for 1p19q codeletion status, there were lower gene expression levels in patients with a codeletion of 1p19q than in patients with a non-codeletion of 1p19q ($p < 0.001$).

## Survival outcomes of *HOXD11* in patients with glioma

The relationship between overall survival and the expression level of *HOXD11* in patients with gliomas was explored by Kaplan–Meier survival analysis. The 749 patients with gliomas included in the study were divided into two groups: "group H" and "group L," based on the median expression level of *HOXD11*. As shown in Fig. 3A, the Overall Survival (OS) was significantly lower in patients with gliomas having high expression levels of *HOXD11* (group H) compared to those with low levels (group L; $p < 0.001$) in CGGA-sequence, which indicated poor prognosis.

To improve the credibility of the data results, we further obtained 666 glioma tissues in the TCGA database. The Kaplan–Meier method was used to verify that the increase of *HOXD11* expression level can indeed reduce the overall survival time of glioma patients (Fig. 3B). To clarify the impact of *HOXD11* expression level on the prognosis of gliomas of different molecular subtypes, we divided the samples into three groups (IDH mutation with 1p19q codeletion, IDH mutation without 1p19q codeletion and IDH wild type) for survival analysis. The results showed that the prognosis of the group with high *HOXD11* expression was significantly worse than that of the group with low *HOXD11* expression among the three molecular subtypes, as shown in Figs. 3C–3E. These results suggest that high expression of *HOXD11* may lead to poor prognosis of patients with glioma.

## High expression of *HOXD11* is an independent risk element in patients with gliomas

To determine whether *HOXD11* could be used as an independent prognostic factor, we performed univariate and multivariate analyses for the date from CGGA-sequence. Univariate analysis was performed by applying Cox regression model (Fig. 4A). The results suggested that high expression levels of *HOXD11* was significantly related to poor prognosis as a risk factor for glioma ($p < 0.001$; HR = 1.550; 95% CI [1.456–1.650]). The PRS type, histology, WHO grade, age and other factors showed similar results. Next, multivariate

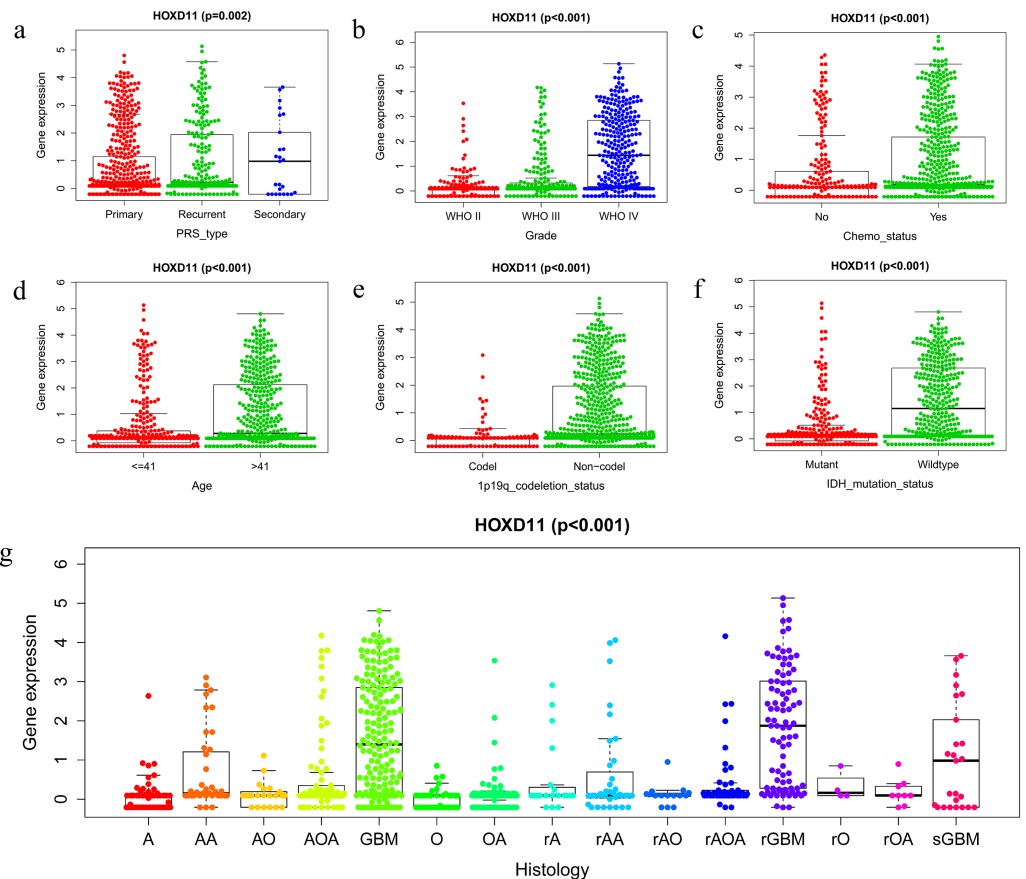

**Figure 2   The relationship between the difference of *HOXD11* expression level and various clinical characteristics.** The various clinical characteristics include (A) PRS type, (B) Grade, (C) Chemo status, (D) Age, (E) 1p19q codeletion status, (F) IDH mutation status, (G) Histology. A: strocytoma; AA: anaplastic astrocytoma; AO: anaplastic oligodendroglioma; AOA: anaplastic oligoastrocytoma; GBM: glioblastoma multiforme; O: oligodendroglioma; OA: oligoastrocytoma; rA: recurrence of strocytoma; rAA: recurrence of anaplastic astrocytoma; rAO: recurrence of anaplastic oligodendroglioma; rAOA: recurrence of anaplastic oligoastrocytoma; rGBM: recurrence of glioblastoma multiforme; rO: recurrence of oligodendroglioma; rOA: recurrence of oligoastrocytoma; sGBM: secondary glioblastoma.

analysis was performed using the Cox regression model (Fig. 4B). High expression of *HOXD11* ($p < 0.001$; HR = 1.156; 95% CI [1.072–1.246]), PRS type ($p < 0.001$; HR = 1.947; 95% CI [1.655–2.291]), and high WHO grade ($p < 0.001$; HR = 2.673; 95% CI [1.955–3.655]) were significantly associated with poor prognosis. Together, the univariate and multivariate analyses results suggest that high expression levels of *HOXD11* may be an independent risk factor for poor prognosis.

## High expression of *HOXD11* has certain evaluation value for glioma prognosis

To clarify the diagnostic value of highly expressed *HOXD11* for glioma prognosis, we plotted the receiver operating characteristic (ROC) curve based on the sequencing data of glioma patients from CGGA and TCGA databases. As shown in Figs. 4C–4D, the AUC

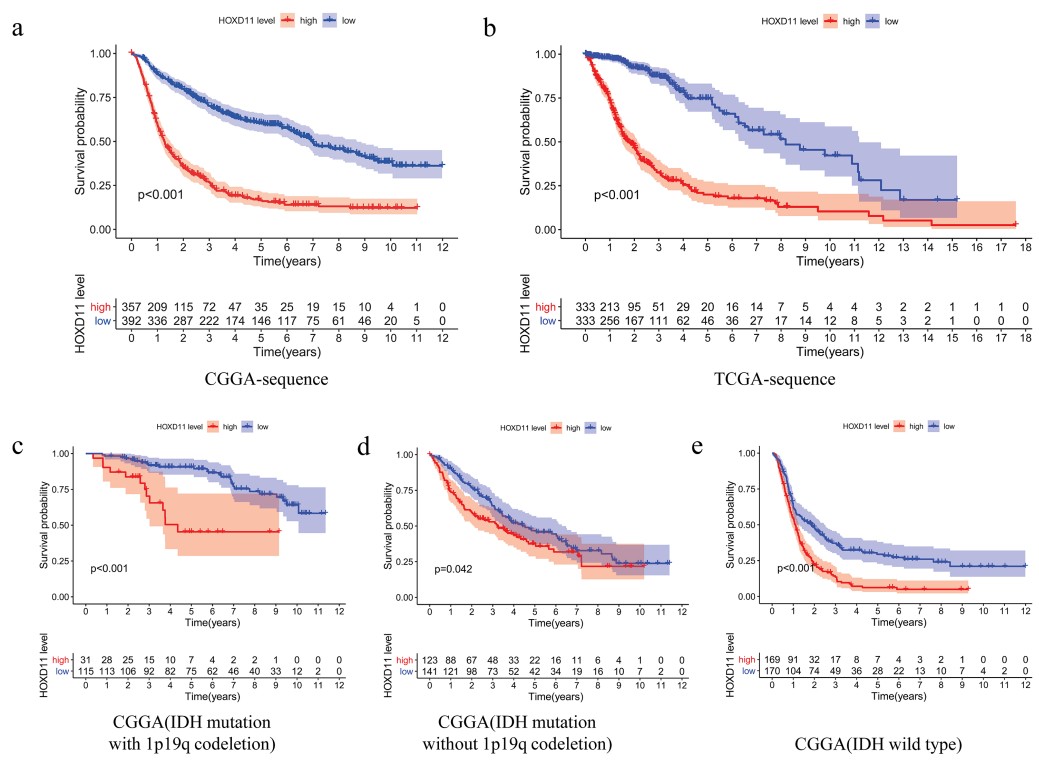

**Figure 3** **The relationship between the difference of the expression level of *HOXD11* and Overall Survival (OS).** (A) CGGA sequence; (B) TCGA sequence; (C) IDH mutation with 1p19q codeletion from CGGA sequence; (D) IDH mutation without 1p19q codeletion from CGGA sequence; (E) IDH wild type from CGGA sequence.

values were greater than 0.700 in each dataset, indicating that high expression of *HOXD11* had certain evaluation value for one-year, three-year and five-year survival rates. These results indicate that *HOXD11* expression level could be related to prognosis in patients with gliomas; moreover, high expression of *HOXD11* may be used for clinical prognosis evaluation of patients with glioma.

## Knockout of *HOXD11* could significantly inhibit the proliferation ability of U251 cells

By analyzing the sequencing data of glioma samples and matching clinical information from multiple sources, the results showed that *HOXD11* could be an independent prognostic factor for glioma and had certain diagnostic value. To verify the effect of *HOXD11* on malignant biological behavior of glioma cell lines, we conducted a series of phenotypic experiments. First, we used siRNA to interfere with the expression level of *HOXD11* in glioma cell U251. Meanwhile, the transfection efficiency was detected by RT-qPCR (Fig. 5A). To determine the effect of *HOXD11* on the proliferation ability of U251 cells, we performed MTT assay. The results showed that the proliferation ability of *HOXD11* knockdown group(*HOXD11*-siRNA group) was significantly reduced compared with the control group(*HOXD11*-NC) (Fig. 5E). Meanwhile, *HOXD11* knockdown also reduced

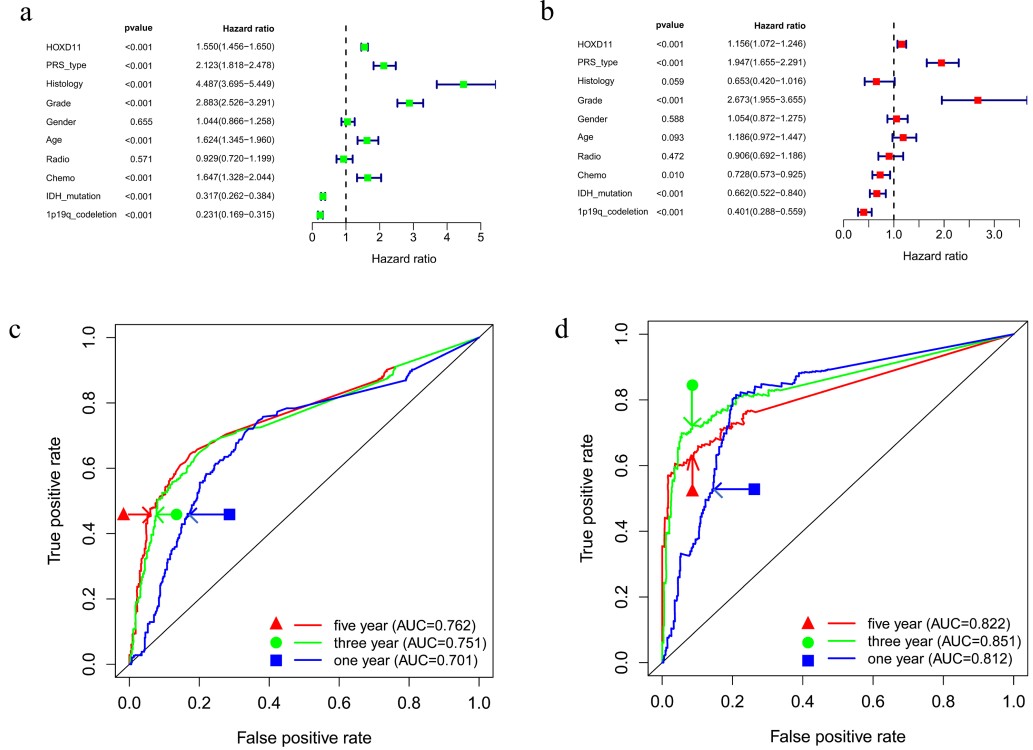

**Figure 4** **Prognostic factors and risk assessment of glioma and the diagnostic value of *HOXD11*.** (A) Univariate regression of prognostic in patients with glioma; (B) Multivariate survival model of prognostic in patients with glioma; (C) The receiver operating characteristic (ROC) curve in CGGA sequence; (D) The receiver operating characteristic (ROC) curve in TCGA sequence.

the ability of cell colony formation (Fig. 5B). Meanwhile, immunofluorescence staining results showed that the expression level of cell proliferation marker KI67 in the knockdown group was significantly lower than that in the control group (Figs. 5C, 5D). In addition, similar results were obtained in the wound-healing assay, and the wound healing ability was also significantly reduced in the *HOXD11* knockdown group (Fig. 5F). Therefore, we believe that highly expressed *HOXD11* is involved in the pathological process of glioma by affecting the proliferation and migration of glioma cells.

## Determination of *HOXD11*-related cellular signaling pathway by GSEA

To clarify how *HOXD11* plays a role in the pathological process of glioma, we performed GSEA analysis to predict the cell signaling pathways that *HOXD11* may participate in. As shown in Table 1 and Fig. 6A, *HOXD11* may be involved in a variety of cancer-related cell signaling pathways such as cell cycle, DNA replication, ECM receptor interaction, and focal adhesion. These results reveal that *HOXD11* may be involved in these cancer-related cell signaling pathway as an oncogene.

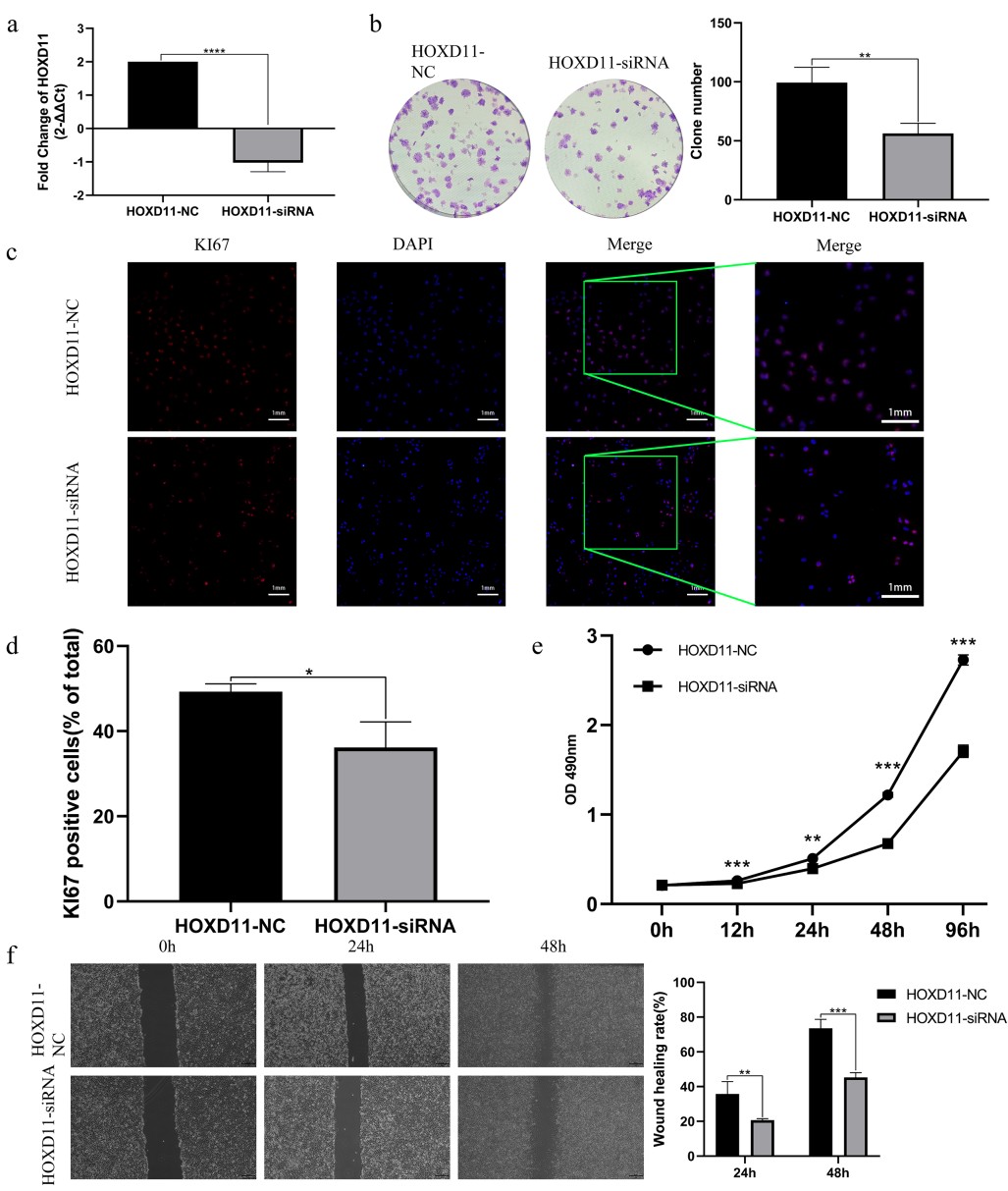

**Figure 5** *HOXD11* **may be involved in promoting the proliferation and invasion of glioma.** (A) RT-qPCR after cell transfection; (B) colony formation assay after cell transfection; (C) KI67 Immunofluorescence staining after cell transfection; (D) statistical graph of KI67 immunofluorescence, $\star$($p < 0.05$); (E) MTT assay at 0 h, 12 h, 24 h, 48 h and 96 h after cell transfection,$\star\star$($p < 0.01$),$\star\star\star$($p < 0.001$);(F) Wound-healing assay at 0 h,24 h,48 h after cell transfection ,$\star\star$ ($p < 0.01$),$\star\star\star$($p < 0.001$).

## High expression of *HOXD11* participates in the pathological process of glioma by regulating cell cycle

Through GSEA analysis, we predicted that highly expressed *HOXD11* may participate in the pathological process of glioma through a variety of cancer-related cell signaling pathways.

**Table 1** The gene set enriches the high *HOXD11* expression phenotype.

| Gene set name | NES | NOM p-val | FDR q-val |
| --- | --- | --- | --- |
| KEGG-CELL CYCLE | 1.728216 | 0.023 | 0.117 |
| KEGG-DNA REPLICATION | 1.6110542 | 0.048 | 0.194 |
| KEGG-ECM RECEPTOR INTERACTION | 1.8454952 | 0.011 | 0.177 |
| KEGG-FOCAL ADHESION | 1.7667084 | 0.013 | 0.118 |

**Notes.**
NES, normalized enrichment score; NOM, nominal; FDR, false discovery rate.
Gene sets with |NES| > 1, NOM *P*-value < 0.05 and FDR *q*-value < 0.25 were considered as significantly enriched.

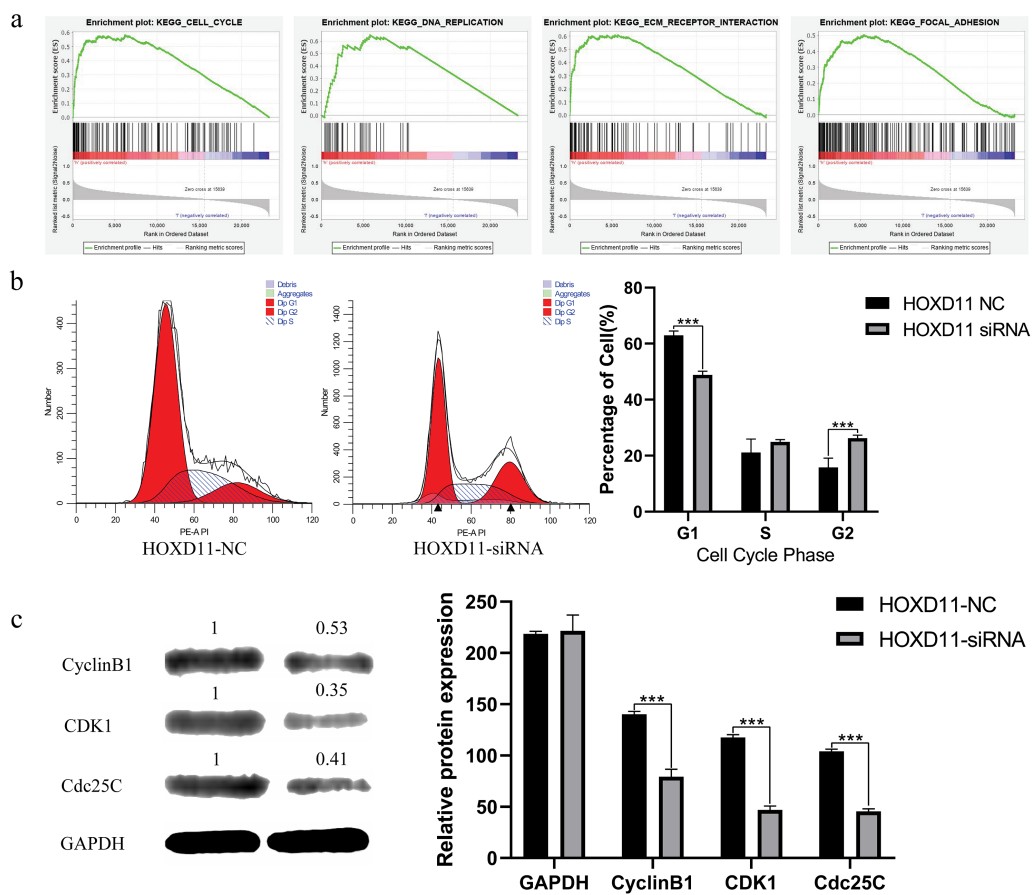

**Figure 6** *HOXD11* may be involved in regulating the G2 phase of the cell cycle in gliomas. (A) Enrichment plots from GSEA. GSEA results showing cell cycle, DNA replication , ECM receptor interactions, focal adhesions differentially enriched in the high expression phenotype of *HOXD11*; (B) Cell cycle-related flow cytometry, *** ($p < 0.001$); (C) The expression level of cell cycle-related protein, GAPDH was used to confirm equal protein loading, *** ($p < 0.001$).

We further chose cell cycle signaling pathway as an object to verify the prediction results. To explore the effect of *HOXD11* on the cell cycle of glioma cell U251, we performed flow cytometry. The results showed that in the *HOXD11* knockdown group, U251 cells were significantly blocked in G2 phase compared with the control group (Fig. 6B). To validate

this result, we performed western blotting. The results showed that the expression levels of cell cycle-related proteins (CCNB1, CDK2 and Cdc25c) were significantly reduced in U251 cells with *HOXD11* knockdown, confirming that *HOXD11* is involved in regulating cell cycle progression (Fig. 6C).

## DISCUSSION

The results of our study indicate that *HOXD11* is highly expressed in glioma samples relative to non-tumor brain tissue. In addition, high expression of *HOXD11* may be associated with poor prognosis in glioma patients. Moreover, ROC analysis verified the diagnostic value of *HOXD11*. These results indicate that *HOXD11* may be used for the treatment, prognosis evaluation and prediction of glioma as a biomarker.

Numerous studies have shown that *HOXD11* is involved in tumor development and that it helps regulate gene expression. *Xu et al. (2019)* showed that abnormal methylation of *HOXD11* may be related to lung cancer pathology. At the same time, studies have suggested that aberrant methylation of *HOXD11* can be used to assess risk stratification and early diagnosis of lung cancer as a biomarker (*Harada et al., 2019*). Studies have suggested that *HOXD11* is highly expressed in Oral Squamous Cell Carcinoma (OSCC) as a homeobox gene (*Rodini et al., 2012*). In addition, *HOXD11* may participate in the progression of laryngeal squamous cell carcinoma by promoting cell proliferation and cell migration (*De Barros et al., 2016*). *HOXD11* may affect the progression of prostate cancer by encoding transcription factors (*Hayano et al., 2016*). Moreover, *HOXD11* may be involved in cell proliferation and angiogenesis-promoting pathways as a cancer candidate gene in hemangioblastoma (*Mehrian-Shai et al., 2016*). Some studies have also found that *HOXD11* can be used as a candidate cancer gene for the diagnosis of breast cancer due to its high methylation levels (*Miyamoto et al., 2005*). Similarly, *Cai et al. (2007)* showed that *HOXD11* can be used as a candidate biomarker in patients with ovarian cancer due to its abnormal methylation. At the same time, studies have explored the role of *HOXD11* in regulating cellular levels in cancer cells. A study by Daniel et al. found that *HOXD11* was significantly expressed in Head and Neck Squamous Cell Carcinoma (HNSCC), and knockdown of *HOXD11* was able to reduce invasion in HNSCC (*Sharpe et al., 2014*). *HOXD11* may enhance cell growth, clonality, and metastatic potential in Ewing sarcoma (*Von Heyking et al., 2016*). Although a number of studies have suggested that *HOXD11* plays a crucial role in cancer progression, the role of *HOXD11* in gliomas has not yet been elucidated.

We found that *HOXD11*, as a novel carcinogenic gene in gliomas, expression levels were significantly higher in gliomas than that in non-tumor brain tissues. This result is completely credible and in line with the rigor of scientific research. Firstly, there are more than a thousand of glioma tissue samples in this result, and the data types contain data from multiple races groups, such as White, Asian and Black or African American. Secondly, the data used for analysis comes from a variety of analysis technologies including RNA sequencing, gene chip technology, RT-qPCR and other technologies, which can complement each other and ensure the authenticity and accuracy of the data. Third, the

Kaplan–Meier method and Cox-regression analysis showed that *HOXD11* could lead to poor prognosis as an independent risk factor. Finally, it is worth emphasizing that the knockdown of *HOXD11* can inhibit the proliferation and migration of U251 glioma cells. Therefore, we have sufficient reason to confirm that *HOXD11* can be a valuable biomarker for targeted treatment of glioma patients.

In order to reveal the mechanism of *HOXD11* leading to poor prognosis in glioma patients, GSEA was used to indirectly reveal the signal pathway activated by *HOXD11*, which indicated that *HOXD11* is significantly enriched in cancer-related cell signaling pathways, such as cell cycle, DNA replication, ECM receptor interaction, and focal adhesion. As one of the most important cell signaling pathways, the cell cycle plays an extremely crucial role in the development of various diseases, especially the progression of cancer. Studies have shown that mutations in genes that regulate the cell cycle may be associated with glioma progression, recurrence, and poor prognosis in patients with gliomas (*Jonsson et al., 2019*). Moreover, *Zhou et al. (2019)* showed that knockdown of Tripartite Motif containing (TRIM44) can affect glioma cell growth by influencing cell cycle regulation. Studies have shown that Bromodomain containing 4 (BRD4) may affect the proliferation and apoptosis of glioma cells through affecting DNA replication in the glioma cell line U251 (*Du et al., 2018*). Focal adhesion plays a crucial role in the development of tumors by affecting cell adhesion and migration (*Paluch, Aspalter & Sixt, 2016*). It has been reported that Methyl Gallate (MG) is able to inhibit the migration ability and activity of glioma cells by inhibiting the formation of focal adhesion (*Lee et al., 2013*). In glioma cells, changes in Glial Fibrillary Acidic Protein (GFAP) can affect the invasiveness of glioma cells by affecting the expression of the Dual-Specificity Phosphatase 4 (DUSP4) in focal adhesions (*Van Bodegraven et al., 2019*). In combination with the above studies, we believe that *HOXD11* may play a role in the aforementioned signaling pathway and thus affect the activity, adhesion, proliferation, and migration of glioma cells.

Although we have incorporated both the data sources of diverse for analysis, which contains Chinese, African American, and Caucasian individuals, so that the differences observed may not be narrowly limited to one ethnicity. Nonetheless, this research still has certain shortcomings to be improved. For example: based on the WHO glioma classification in 2016, more attention has been paid to the molecular typing of glioma. However, most of our data were obtained from public databases which may not necessarily contain information on the molecular typing of gliomas. Therefore, if conditions permit, we may develop more detailed research programs related to the molecular typing of gliomas. As another limitation, the mechanism whereby high expression of *HOXD11* leads to poor prognosis in glioma patients was indirectly revealed by functional enrichment analysis via GSEA. We only confirmed that *HOXD11* knockdown could block U251 cells to G2 phase by flow cytometry and western blotting. Therefore, other specific mechanisms of *HOXD11* affecting glioma need to be further verified by other research. This study only provides a clue to lead more researchers to pay attention to the role of *HOXD11* in glioma.

## CONCLUSIONS

This study found that compared with normal controls, the expression level of *HOXD11* was higher in glioma samples at the cell and tissue level, and may be related to the poor overall survival of patients with glioma. In addition, *HOXD11* may be involved in cancer-related cell signaling pathways, such as cell cycle, DNA replication, focal adhesion, and ECM receptor interaction, and thereby affect the malignant progression of glioma by participating in the regulation of the cell cycle. Therefore, *HOXD11* could be used as a candidate biomarker for molecular therapy and prognostic evaluation of patients with glioma.

## ACKNOWLEDGEMENTS

We appreciate the support of Affiliated of Henan Provincial People's Hospital.

### Funding

This work was also supported by the Central Plains Thousand Talents Plan of Henan Province under Grant (No. ZYQR20191212). The funders had no role in study design, data collection and analysis, decision to publish, or preparation of the manuscript.

### Grant Disclosures

The following grant information was disclosed by the authors:
The Central Plains Thousand Talents Plan of Henan Province: ZYQR20191212.

### Competing Interests

The authors declare there are no competing interests.

### Author Contributions

- Jialin Wang and Zhendong Liu conceived and designed the experiments, performed the experiments, prepared figures and/or tables, and approved the final draft.
- Cheng Zhang, Binfeng Liu, Xiaoyu Lian, Zhishuai Ren, Wang Zhang, Yanbiao Wang, Bo Zhang and Bo Pang analyzed the data, authored or reviewed drafts of the paper, and approved the final draft.
- Hongbo Wang and Ang Li performed the experiments, authored or reviewed drafts of the paper, and approved the final draft.
- Yanzheng Gao conceived and designed the experiments, performed the experiments, prepared figures and/or tables, and approved the final draft.

### Human Ethics

The following information was supplied relating to ethical approvals (i.e., approving body and any reference numbers):

This study has been approved by the Institutional Review Board of Zhengzhou University.The use of patient samples conformed to the declaration of Helsinki.

## Data Availability

Raw data, including the relevant clinical information of nine glioma tissue samples are available, are available in the Supplemental Files.

## Supplemental Information

Supplemental information for this article can be found online at http://dx.doi.org/10.7717/peerj.10820#supplemental-information.

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
