# Peer review of "Abnormal expression of HOXD11 promotes the malignant behavior of glioma cells and leads to poor prognosis of glioma patients"

_PeerJ, doi:10.7717/peerj.10820_

## Round 0.1 · original submission · Minor Revisions

While the study and validity of the findings are sound, addressing the reviewers comments would greatly help in improving the manuscript.

Reviewer 1 ·

Basic reporting

As a reviewer I am not able to verify references for SVA technique used in line 133. Removing batch effects is important when working with multiple sources of data hence referencing this technique would be very useful to the future readers of this manuscript.

In line 134 & 270 authors have not stated what metric (ex: median) have been used to classify the expression levels of HOXD11 into high, low.

For figure 1b, 1c- authors have not stated what type of expression levels they have used. (ex: fpkm, log transformed, z scores etc). Please clarify this.

In line 254- is it age at sample collection or age at diagnosis – please clarify this variable description.


In line 376- can you reference/cite appropriate racial classification? As a reviewer I am not able to verify if there are any official race classifications (ex- yellow) as described here.

Experimental design

no comment

Validity of the findings

no comment

Reviewer 2 ·

Basic reporting

Writing style can be improved a little. Specific examples provided in comment below.

Experimental design

No comment

Validity of the findings

The findings appear valid and well-substantiated.

Additional comments

The authors have performed a thorough evaluation of the expression of HOXD11 in glioma across multiple cohorts. They used GSEA pathway-based analysis to establish the role of HOXD11 as an oncogene in glioma. They further validated the effect of HOXD11 on cell cycle experimentally using flow cytometry and western blotting.
While genes of the HOX family including HOXD11 are known to play a role in the development of a variety of cancers, the role of HOXD11 has not been specifically established in glioma.
Some comments below:
1. The y-axis in figure 1a makes it tricky to see the expression patterns. Cutting off the y-axis to exclude extreme outliers would help more clearly focus on the expression patterns in individual tumor types.
2. In the breakdown of expression by subtype (figure 2g), the x-axis legend is not clear. Detailed subtypes can be added in the legend.
3. Are the survival analysis from CGGA (figure 3) adjusted for age, gender and tumor stage? Have the authors explored the association of HOXD11 with MGMT status?

The writing can be improved to enhance clarity and avoid redundancies. Some suggestions are below:
1. In line number 26, clinical department of which institution? How many samples?
2. It would be helpful if the authors could provide number of samples from each study in abstract or introduction to easily understand the scope – it is currently buried in methods section.
3. The description of TCGA in lines 64-70 can be shortened as this is widely accepted across the field.
4. There is a typo in line 74 – CGGA (as opposed to CCGA)
5. Data description in lines 374-376 can improved.

Reviewer 3 ·

Basic reporting

1. The description of TCGA in line 64, 117 is redundant
2. There is a grammatical error in line 239
3. The Figure text is a little unreadable- can it be made a more clear.
4. The fig 5c is not very visible unless you zoom in- could that be modified?
5. For fig 6c, can the authors provide an uncropped image of the blot

Experimental design

The authors have conducted a thorough investigation of HOXD11 in glioma prognosis, very well utilizing the public datasets.

1. The authors used publicly available datasets to show the higher expression of HOXD11 in tumor vs. normal- were the datasets preprocessed before analysis. Did the authors consider any possible covariate in the dataset.
2. In fig 1b- a few samples from normal were in the range above the mean of tumor sample expression- what was different about these two samples?
3. In the survival analysis, the IDH wildtype does have significant difference between High and low HOXD11, however this difference seem to be minimal with IDH mutation without 1p19q codeletion. Did the authors consider to condition on the IDH mutation without 1p19q codeletion?

Validity of the findings

HOXD11 plays a significant role cancer, have the authors checked for literature describing a similar role in any other type of cancer.

Additional comments

The authors in the manuscript show the role of HOXD11 through bioinformatic analysis using publicly available datasets and through a series of experiments. The authors very efficiently use the available datasets to show the association of high expression of HOXD11 with age, WHO grade, chemotherapy status, histological type and 1p19q codeletion data and isocitrate dehydrogenase (IDH) mutation. They further validate the pathway predicted by GSEA with experiments proving knockdown HOXD11 inhibits proliferation, invasion and to prevent cell cycle progression.

---

## Round 0.2 · accepted · Accept

Thank you for carefully addressing each of the reviewers' comments. The manuscript has improved significantly and can be accepted for publication.